# Influenza Vaccination Uptake and Prognostic Factors among Health Professionals in Italy: Results from the Nationwide Surveillance PASSI 2015–2018

**DOI:** 10.3390/vaccines11071223

**Published:** 2023-07-10

**Authors:** Valentina Minardi, Rosaria Gallo, Valentina Possenti, Benedetta Contoli, Davide Di Fonzo, Elvira D’Andrea, Maria Masocco

**Affiliations:** 1National Centre for Disease Prevention and Health Promotion, Italian National Institute of Health, 00161 Rome, Italy; valentina.minardi@iss.it (V.M.); benedetta.contoli@iss.it (B.C.); maria.masocco@iss.it (M.M.); 2Primary Healthcare Unit, Health District 9, Local Health Unit Roma 2, 00159 Rome, Italy; rosariagallo.cs@gmail.com; 3PhD Course Advances in Infectious Diseases, Microbiology, Legal Medicine and Public Health Sciences, Sapienza University of Rome, 00185 Rome, Italy; 4Local Health Unit 2 Liguria, 17100 Savona, Italy; difonzo.davide@gmail.com; 5Division of Pharmacoepidemiology and Pharmacoeconomics, Brigham and Women’s Hospital, Harvard Medical School, Boston, MA 02115, USA; elvira.dandrea@gmail.com

**Keywords:** influenza vaccination, health personnel, health care worker, surveillance system, epidemiology, prevention, public health

## Abstract

(1) Influenza causes a significant health and socio-economic burden every year, and health personnel (HP) are at higher risk of exposure to respiratory pathogens than the general population. (2) The study’s purpose was to describe and compare influenza vaccine uptake and its prognostic factors among Medical Doctors (MDs) and Non-Medical Health Personnel (NMHP) vs. Non-HP (NHP). We analyzed 2014–2018 data (*n* = 105,608) from the Italian Behavioral Risk Factor Surveillance System PASSI that, since 2008, has been collecting health-related information continuously in sampled adults. (3) MDs and NMHP represented, respectively, 1.1% and 4.6% of the sample. Among HP, 22.8% (CI 19.8–26.1%) of MDs and 8.5% (CI 7.5–9.5%) of NMHP reported to have been vaccinated vs. 6.3% (CI 6.1–6.5%) in NHP. This difference is confirmed in the three categories (MDs, NMHP, NHP), even more across age groups: in 18–34 yy, respectively, 9.9%, 4.4%, 3.4% vs. 28.4%, 13.9%, 10.6% in 50–64 yy. PASSI surveillance shows an increasing influenza vaccination uptake over time, especially among MDs (22.2% in 2014 vs. 30.5% in 2018). (4) Despite such an increase, especially among younger HP, influenza vaccination uptake is low. Even more under pandemic scenarios, these figures represent key information to address effective strategies for disease prevention and health promotion.

## 1. Introduction

Influenza (flu) is a highly contagious viral disease with global circulation. The World Health Organization estimated that epidemics of seasonal influenza interest about 5–15% of the population worldwide annually, causing up to five million cases of severe illness and 650,000 respiratory deaths [1]. Influenza can cause mild to severe illness, with hospitalization and death occurring most frequently among high-risk groups such as older adults, individuals with underlying chronic health conditions, pregnant women, and young children [2].

Medical visits and hospitalizations associated with influenza facilitate the transmission of the infection to healthcare professionals. Due to increased exposure to influenza-related illness, healthcare workers are at high risk of acquiring influenza and infecting colleagues and patients under their care. This contributes to sustain and perpetuate the risk of direct transmission to vulnerable individuals, and also trigger flu epidemics within a hospital or other healthcare settings. Thus, healthcare professionals represent an important target population for vaccination against influenza as a vaccine-preventable disease (VPD) [3]. Several studies demonstrate that healthcare flu vaccinations are associated with reductions in all-cause mortality among hospitalized vulnerable patients, influenza-like illness, and hospitalizations for influenza. In addition to maintaining essential healthcare delivery during outbreaks and acting as a barrier against the spread of infections [4], vaccinated healthcare professionals contribute also to effectively addressing concerns, mistrust, and misconceptions among the general population because regarded as reliable and trusted sources of information [5,6].

Despite immunization of healthcare professionals is recommended in many countries [7], its uptake remains generally low or unclear [8]. In Italy, the National Immunization Prevention Plan (PNPV) provides guidance over the years for the regional vaccine and immunization enterprise as the nation seeks to reduce and eventually eliminate VPDs. As a consequence of the COVID-19 pandemic [9], the Ministry of Health (MoH) has issued specific recommendations to health professionals for several types of vaccination as essential for the prevention and control of infection-related diseases, such as hepatitis B, pertussis, measles, mumps, rubella, varicella, tuberculosis, and also influenza [4,10,11]. Although the annual flu vaccination of healthcare professionals is strongly recommended by the PNPV, the success of vaccination programs and campaigns targeting this group is unclear, as coverage data are not readily accessible and only limited to local communities or a few hospitals [10,12]. Limited access to immunization data is due to the lack of a national vaccine registry, which is a major goal of the current PNPV 2023–2025.

This study aims to provide national estimates on influenza vaccination uptake among health professionals by analyzing the 2014–2018 data from the Italian Behavioral Risk Factor Surveillance System (BRFSS) PASSI (Progressi delle Aziende Sanitarie per la Salute in Italia—Progresses in assessing population health in Italy). Further, we investigated the association between demographic, medical, and socioeconomic factors and adherence to immunization programs for influenza among healthcare providers.

## 2. Materials and Methods

### 2.1. Study Design and Data Source

We performed a cross-sectional study by using data generated by PASSI, an ongoing nationwide surveillance system centrally coordinated by the National Institute of Health (Istituto Superiore di Sanità, ISS) and committed by the National Center for Disease Prevention and Control of the Italian MoH [13].

PASSI collects data on many of the behaviors and medical conditions that could influence the health status of adult people residing in Italy, as per the domains and indicators encompassed in the National Prevention Plan [14]. The target population consists of individuals aged 18 to 69 years residing in the area of a Local Health Unit (LHU) that participates in the surveillance program. The criteria for eligibility include the availability of a telephone number, both landline and cell phone numbers, and the capacity to have a phone interview with specially LHU-trained personnel who administer a standardized questionnaire on the main behavioral health-related risk factors. In each LHU, a random sample is drawn on a monthly basis from the residents’ list, stratified by sex and age (18–34, 35–49, and 50–69), proportionally to the size of the respective strata in the general population. Data gathered in each LHU are then merged, weighted, and analyzed to obtain national and regional estimates. The sample is representative of the adult population residing in Italy. A detailed description of the validated survey design and random sampling procedures is available elsewhere [15,16].

#### Sampling Procedure

A total of 154,187 eligible participants was identified in the period 2014–2018; and the sample was representative of the population living in Italy (response rate to the survey: 81%; LHU participation rate to PASSI nationwide: 110 out of 121 LHUs). Overall, 105,598 participants who provided information on their flu vaccination status and reported being employed at the time of the interview were selected (Figure 1). We then classified participants into three groups based on the information collected on “work sector” and “type of job”: 5979 individuals working as Health Personnel (HP), 1697 personnel employed in the health sector without health duty, and 97,922 workers in other sectors. Among HP, 1178 were Medical Doctors (MDs) and 4801 were Non-Medical Health Personnel (NMHP), thus including nursing, midwifery, and other health-related professions. In this analysis, we did not include people who, although working in the health sector, did not practice any health profession (Figure 1).

### 2.2. Outcome and Covariate Variables

The outcome was adherence to seasonal influenza vaccination, which was obtained by categorizing the answers to the question “In the past 12 months, have you had seasonal flu vaccination?” into a dichotomous variable (yes vs. no/I do not know/I do not remember”).

Covariate variables were socio-demographic characteristics and health-related conditions. The formers include gender, age (18–34; 35–49; 50–64), economic difficulties in making ends meet by the available financial resources (many or some; not at all), educational attainment (low as none or elementary or middle school; high as high school or university), macro-area of residence (North; Center; South of Italy), urbanization degree, marital status (married yes vs. no), and nationality (Italian vs. others). Specifically, regarding the urbanization degree, ISTAT classifies the Italian municipalities [17] by dividing the territory into 1 km^2^ cell clusters, considering the two criteria of geographic contiguity and minimum population threshold, and matching the resulting areas with the administrative boundaries of the municipalities. These last are categorized as per population density level: high —with at least 50% of the population living in densely populated areas; intermediate—with <50% of the population living in rural areas and <50% in densely populated areas; low—where >50% of the population falls into rural areas.

Health-related conditions included the presence of at least one chronic disease, obesity, tobacco smoking, and self-perceived health status. The former is investigated by asking the interviewees if a physician has ever diagnosed or confirmed any of the following: kidney failure, chronic bronchitis, emphysema, respiratory failure, bronchial asthma, stroke or cerebral ischemia, diabetes, myocardial infarction, ischemic heart or coronary artery disease, other heart diseases, cancers (including leukemia and lymphoma), chronic liver disease or cirrhosis.

### 2.3. Statistical Analysis

Prevalence of adherence to seasonal flu vaccination was calculated stratified by type of healthcare employee, Medical Doctors (MDs) and Non-Medical Health Personnel (NMHP), and for the general population, excluding health professionals (NHP). Furthermore, results were provided according to all covariate variables mentioned above.

Bivariate and multivariate logistic regression analysis was used to evaluate the association between vaccination adherence outcome and sociodemographic and medical factors such as age, gender, educational level, economic difficulties, macro-area of residence, and the presence of at least one chronic disease. Odds ratio (OR) estimates, crude and adjusted, 95% confidence interval (CI), and *p*-value were reported. We used univariate logistic analysis by subgroup to test the association of socio-demographic characteristics and health-related conditions with adherence to the seasonal influenza vaccination campaigns. We then conducted a multivariate regression logistic model by a subgroup that excluded variables with *p* > 0.20 in the univariate analysis.

STATA version 17.0 was used for all. Throughout the paper, *p*  <  0.05 is regarded as significant.

## 3. Results

### 3.1. Socio-Demographic Characteristics and Health Conditions

Socio-demographic characteristics and health conditions by professional category (MDs, NMHP, and NHP) are shown in Table 1.

Among NMHP, females were more represented (73.9%) than in the other two groups [MDs (47.1%) and NHP (42.7%)].Almost half of the MDs were between 50–64 years and had the highest education level compared to the other groups (89.9% of NMHP vs. 71.5% of NHP had a university or high school diploma), holding at least a university degree. They also reported fewer economic difficulties.Most respondents resided in municipalities with an intermediate urbanization degree. MDs lived more frequently in municipalities with high population density (48.9%) than NMHP (35.1%) or NHP (33.7%).MDs were less likely to smoke (16.4%) compared with other categories (25.4% of NMHP and 28.7% of NHP).The general population (9.4%) was more likely to have obesity than NMHP (7.9%) or MDs (6.4%).

From 2014 to 2018, we detected an overall increase in the adherence rate to the seasonal influenza vaccination across the three groups, with a rise from 2016. The adherence rate increased by over 25% among MDs (from 22.2% in 2014 to 30.5% in 2018), although no statistical significance due to the small sample size. A positive trend was also observed in the NHP group (Figure 2).

### 3.2. Adherence to the Flu Vaccination Campaign by Socio-Demographic Characteristics and Health Conditions

In the period 2014–2018, the adherence rate to the seasonal influenza vaccination of MDs was significantly higher (22.8%) than that observed among NMHP (8.5%), who still reported a higher adherence than the general population (6.3%) (Table 2).

Adherence was lower in females in comparison to males (respectively, 21.3% vs. 24.1% for MDs, and 7.8% vs. 10.2% for NMHP), and increased by age across all three categories, particularly among MDs (28.4% vs. 9.9% in doctors aged <35 years). MDs’ adherence to vaccination was higher in the North (25.7%) and in the South (22.7%), than in the Center (17.9%) of Italy. Among NMHP and NHP, a higher adherence was observed in the South (9.6% and 6.9%, respectively) compared to the North (7.9% and 5.7%, respectively). MDs living in high-urbanized municipalities were reported to be less vaccinated than those living in areas with a lower degree of urbanization. No substantial difference is observed among NMHP or NHP. Married participants seemed to have better adherence to flu vaccination than those unmarried, although the proportions were not significantly different. A difference in adherence has been observed in Italian vs. other NMHP (Table 2).

In all three groups, there was a higher adherence to vaccination in people with at least one chronic disease (*vs* none): MDs 36.3% vs. 20.8%; NMHP 13.9% vs. 7.5%; NHP 17.6% vs. 4.4%. Likewise, a similar occurrence was observed in whom reported obesity; on the contrary, tobacco smoking resulted not to be associated with the adherence levels in HP. Finally, independently from the belonging category, those who perceived lower health reported to vaccinate more against seasonal influenza (Table 3).

In the multivariate analysis, being an MD was associated with a higher adherence rate to flu vaccination compared to NMHP and the general population after controlling for socio-demographic characteristics and risk factors.

In the multivariate analysis, adherence of MDs to the seasonal influenza vaccination was significantly associated with age (AdjOR = 3.14 in 50–64 vs. under 35 years), geographic area (AdjOR= 0.64 Central vs. Northern Italy), and having at least one chronic disease (AdjOR = 1.85 at least one vs. none). Among NMHP, adherence was instead associated with gender (AdjOR = 0.74 in women vs. men), age (AdjOR = 3.30 in 50–64 NMHP vs. under 35 years) as well as having at least one chronic disease (AdjOR = 1.73 at least one vs. none). Adherence to vaccination of NHP was associated with males, older age, residing in Central and Southern Italy, living in intermediate and low population density municipalities, and having at least one chronic disease (AdjOR = 3.49 at least one, compared to none). Smokers and participants who were referred to be in good health status were significantly less vaccinated (Table 4).

## 4. Discussion

### 4.1. Critical Reading of the Results

This study aimed to estimate the rate of influenza vaccine uptake and its prognostic factors among MDs, NMHP, and NHP in Italy by using data gathered through a large population-based surveillance system in the period 2014–2018.

Overall, adherence to flu vaccination among HP increased over time but remains still poor or suboptimal. Although HP are strongly recommended to receive an annual flu vaccination due to their high exposure risk, at the end of 2018, only 30.5% of MDs and 9.6% of NMHPs referred to be vaccinated against influenza. Adherence rates were higher in HP compared to those who do not work in healthcare. MDs were thrice likely to receive an annual flu vaccination compared to NMHP, also after controlling for socio-demographic characteristics and risk factors.

The differences in vaccination coverage observed among different areas (better rates in the North and the South than in the Center) may reflect indirectly the regional policies. When considering prevalences by age in the different HP categories, MDs show higher rates than NMHP in all age groups. However, across all groups, older participants were more likely to report having received a flu vaccine compared to the younger ones. The adherence rate was positively correlated with age, thus observing the lowest proportion of vaccinated among the 18–34-year-old subgroup and the highest among participants with 50–64 years old. This difference between older and younger cohorts, defined as the ‘vaccine confidence gap’, is consistent across many European countries, and is raising main concerns among policymakers since it is widening over time [18]. It is paramount to educate the young population groups about the safety and effectiveness of vaccines in order to increase immunization acceptance in the future.

#### 4.1.1. The Literature Appraisal on Influenza Vaccination Uptake and Its Main Determinants

We conducted scoping research on the literature of seasonal influenza vaccination uptake among health professionals in Italy, in the period 1990–2022, including systematic reviews on the topic (Appendix A). According to the studies we reviewed, the overall adherence rate is quite low. In fact, reported uptake ranges from under 10% to around 20%, with no significant changes since 1990. Lower vaccine acceptance levels are reported in females and nurses (considering the high female-to-male ratio between nurses, some collinearity is to be expected). The main motivations of vaccine acceptance among HP seem to be egoistical: most of those who accept seasonal flu vaccination do so to protect themselves and their family, with a positive association between self-concern and older age and comorbidities. A lesser role is played by altruistic motivation—i.e., willingness to protect the general population and to improve patients’ safety. Furthermore, vaccine uptake is also associated with trust in vaccines and recommending the flu vaccination to patients, as to be expected.

Refusal of vaccination was mostly motivated by low-risk perception; other negative associations were seen with the denial of the social benefit of influenza vaccination, low social pressure, lack of perceived behavioral control, a negative attitude toward vaccines, not having been previously vaccinated against influenza or not having previously had influenza, lack of adequate influenza-specific knowledge, lack of access to vaccination facilities, and socio-demographic variables. Therefore, the path to improving influenza vaccination uptake among HP in Italy is ridden with several challenges and ethical issues [8,19].

As recalled in the Introduction, increasing influenza vaccination coverage among healthcare workers would mean also reducing the risk of potential nosocomial influenza-like illness in patients hospitalized in acute hospitals [3].

#### 4.1.2. Interventions and Application of Proven Effective Policies

Systematic reviews of randomized controlled trials investigating the effectiveness of interventions for improving vaccine uptake among HP found that combined strategies were more effective than isolated approaches. High-quality studies on specific motivations leading to lower vaccine confidence would help policymakers and stakeholders shape evidence-based initiatives and programs to improve vaccination coverage and the control of influenza through the correct application of guidelines on prevention [7,8]. On the other hand, studies that evaluated the effectiveness of multicomponent interventions to increase vaccination coverage found only minimal to moderate increases in uptake levels [12]. For example, a group of different studies retrieved good effectiveness from on-site vaccination sessions [20,21]. Currently, according to the law that regulates safety in the workplace, health surveillance for biological agents and therefore relative vaccination refers to the specific risk factors, whether they are potential or deliberate, and there is no specific reference to flu vaccination. However, this law also encourages information and awareness-raising activities for workers in the vaccination field.

Nevertheless, both the PNPV [9] and the Recommendations for the prevention and control of influenza 2021–2021 have identified health professionals as target groups for influenza vaccination. The interventions that make it “complex and traceable” flu vaccination refusal increase adherence to this type of vaccination. The results show that launched vaccination campaigns have not contributed to an increase in the rate of adherence by health practitioners [19].

As also recalled in the Introduction, the national vaccine registry should ensure the correct assessment of vaccine coverage and the monitoring of vaccine programs implementation at a national level. The PNPV 2017–2019 illustrates functionalities and required datasets which Regions must deliver to inform the regional vaccine registries and to correctly feed the national registry. So far, in Italy, vaccination coverage data are not readily accessible and only limited to local or a few communities [9].

In the United States, mandatory influenza vaccination of HP was introduced more than a decade ago with excellent results [5]. Thus, mandatory policies are currently under debate in several countries. From the findings of a survey among the Vaccination Service employees in the Apulia Region, three respondents out of four stated that mandatory vaccination should be retained [22]. Reviews which also include meta-regression analysis about evaluating interventions to increase seasonal influenza vaccination coverage in HP show that mandatory vaccination results to be the most effective intervention component, followed by “soft” mandates such as declination statements, increased awareness, and increased access. Whereas for incentives the difference was not significant and for education no effect was observed, such evidence claims that effective alternatives to mandatory HP influenza vaccination do exist, and need to be further explored according to context-related specificity [23].

Some studies suggest an expressed need from HP for more education or counselling. Still, despite provided educational programs generating an improvement in vaccination adherence in the engaged HP, the resulting increased coverage is in any way lower than recommended to reduce influenza spread in healthcare contexts such as hospitals. Thus, specific training alone may play a role in improving influenza vaccination adherence, but integration with broader public health measures is envisaged as well [24].

Other kinds of intervention concern participative strategies such as forum theater that does represent an innovative solution to increasing HP awareness of the importance of flu vaccination and could positively impact their adherence to vaccination recommendations. The forum theater methodology can also be a meaningful teaching tool to health professionals for their learning about and changing attitudes toward other clinics and public health issues [25].

#### 4.1.3. Impact of Pandemics

The COVID-19 epidemic may have had a significant impact on the matter of VPDs. For instance, in the 2020–2021 flu season, a pretty higher prevalence of vaccinated HP was observed in Italy [26] or altruistic motivations such as protecting the general population and safety of patients, usually considered less important as vaccination drivers, were much higher accounted during the COVID pandemic. Indeed, it has even represented a watershed also in the matter of mandating influenza vaccination to HP in Italy; in fact, the Law Decree of 01 April 2021 establishes mandatory vaccination against COVID for healthcare workers. Although mandatory is an undesirable modality for operators in the health area, those measures, which are necessary in emergency contexts, can increase awareness in the category. Even more after the COVID-19 pandemic, the relevance of seasonal influenza vaccination uptake among HP is highly related to the key role that they play in representing a behavioral model to their patients, having such an enormous potential to contribute to better vaccine confidence among the general population and its specific subgroups. Considered one of the most trusted sources of information about vaccination, HP are crucial in motivating individuals to become vaccinated. The Vaccine Confidence Project found vaccine confidence among HP across all 27 EU member states to be universally high and stable between 2018 and 2020, with a large perception increase particularly towards the seasonal influenza vaccine [18].

In Italy, in 2022, the confidence level towards flu vaccine achieved 100% on safety, relevance, effectiveness, and 97% on compatibility with beliefs. Furthermore, about the likelihood to recommend the flu vaccine 98% would do it to patients and 88% to pregnant women [27]. In turn, data from the Italian BRFSS PASSI d’Argento (on the elderly population) show that four out of five older individuals referred to have been advised by the general practitioner to vaccinate against flu: this figure remains high as before the COVID-19 pandemic [28].

### 4.2. Strengths and Limitations

This research is the first nationwide cross-sectional study on influenza vaccination uptake among healthcare professionals in Italy. PASSI is an ongoing surveillance system and can monitor how this behavior and prognostic factors change over time, even under emergency occurrences such as the COVID-19 pandemic [29]. PASSI relies on large sample sizes and proven reliability and validity because data aggregated over time gives good solidity to the results, but it does not refer to any European network, thus comparisons with the same data from other Countries are not possible.

This study has some limitations mainly attributable to the self-reporting nature of data collection. Except for the main demographic characteristics such as sex, age, and residence, data cannot be validated with objective measures, and the information released can be somehow biased, generating underestimations or overestimations [30]. Vaccination uptake may be affected by bias for social desirability, and misclassification or recall bias could occur on vaccine compliance, as referring to infrequent behavior. Self-reported data show in any way robust consistency and sensitivity [31], even because the PASSI interviewers are specially trained healthcare workers from the LHU Prevention Departments and, over the years, a capillary surveillance network has been growing on local territory [15]. As described in the Methods, the PASSI design and sampling procedures ensure both low nonresponse rates and much broader coverage than other similar surveys (e.g., the American BRFSS), and the final response rate is goodly assessed [16].

Finally, the health professionals’ sub-sample is reconstructed retrospectively [32] and no data on the specific healthcare working setting are available in PASSI.

## 5. Conclusions

In Italy, the adherence rates to flu vaccine among HP are still too low. This is particularly problematic considering that vaccinating HP can effectively limit nosocomial infections, thus influenza-related illness, deaths, and outbreaks in the healthcare setting. It is crucial to investigate the reasons for the low uptake of influenza vaccine among HP and conduct rigorous immunization campaigns targeting this group of workers. The adherence is even lower among young HP. Again, follow-up investigations in low confidence among younger age groups of HP may be warranted to understand specific barriers to vaccination uptake that is of prime importance if we are to increase immunization uptake rates to sufficient levels in the next future.

Because vaccine confidence is influenced by several external factors and can change in light of developments over time, it is difficult to determine if findings from this study represent short-term fluctuating and reversible trends or more permanent shifts. Continued monitoring of trends in the general population and health professionals, in particular, is vital to identify early warning signals of confidence losses. Moving towards more robust sub-national monitoring and evaluation of trends will allow for insights into spatial heterogeneities in confidence which may provide more local insights into potential vaccination policies and interventions. Consistent monitoring of confidence levels, their causes and consequences, can help shape our understanding of and ability to predict future trends as well as inform interventions to mitigate the negative outcomes of vaccine hesitancy and resulting decreases in vaccine uptake.

These insights can constructively help the identification of effective strategies to significantly increase vaccine coverage in order to decrease the risk of nosocomial infections, prevent transmission to patients, and reduce indirect costs related to HP absenteeism due to illness. Even more under pandemic scenarios, given the role of professionals as public health promoters in society, these figures represent key information to address effective strategies in the matter of VPD prevention.

Joint efforts with occupational medicine, workplace prevention, and health promotion interventions are needed to develop a culture of the protection of public health and patients. In this framework, the introduction of compulsion would be more to be understood as a requirement of safe work for the worker and patients, than as a coercive action. Therefore, specially designed educational programs should be set up to make health professionals interiorize the value and worth of voluntary compared to mandatory vaccination, as well as why high vaccination rates do not have to depend on compulsion.

## Figures and Tables

**Figure 1 vaccines-11-01223-f001:**
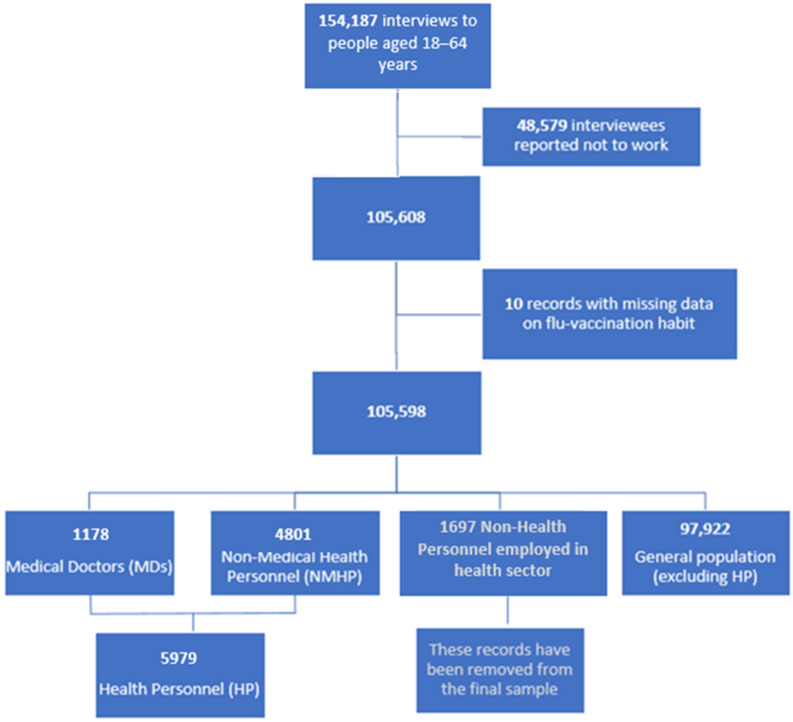
Flow chart describing the subsample selection procedure from the working-age respondents to the surveillance system PASSI interview: Health Personnel (HP) including Medical Doctors (MDs) and Non-Medical Health Personnel (NMHP), 2014–2018 (*n* = 169,678).

**Figure 2 vaccines-11-01223-f002:**
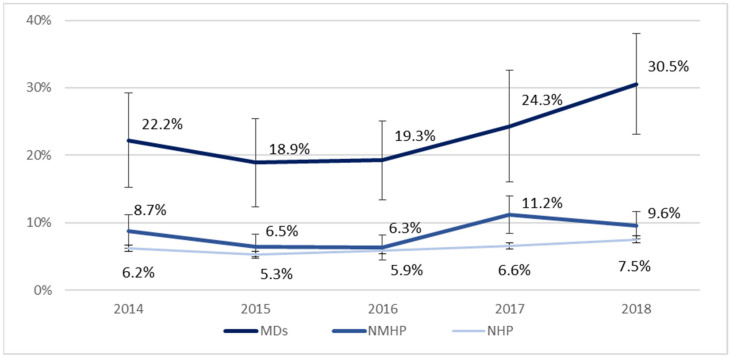
Trends of adherence to seasonal influenza vaccination campaigns in the three category samples: Medical Doctors (MDs), Non-Medical Health Personnel (NMHP), and General population excluding HP (NHP). PASSI 2014–2018 (*n* = 103,901).

**Table 1 vaccines-11-01223-t001:** Socio-demographic characteristics and health conditions of the three category samples: Medical Doctors (MDs), Non-Medical Health Personnel (NMHP), and General population excluding HP (NHP). PASSI 2014–2018 (*n* = 103,901).

	Healthcare Personnel (HP)				General Population (Excluding HP) (n = 97,922)	
	Medical Doctors (MDs) (n = 1178)		Non-Medical Health Personnel (NMHP) (n = 4801)			
	%	(IC 95%)	%	IC 95%	%	IC 95%
Total	1.1	–	4.6	–	94.3	–
Gender						
Men	52.9	(50.0–55.7)	26.1	(24.9–27.4)	57.3	(57.0–57.6)
Women	47.1	(44.3–50.0)	73.9	(72.6–75.1)	42.7	(42.4–43.0)
Age group						
18–34	17.7	(15.7–20.0)	21.5	(20.4–22.7)	24.9	(24.6–25.1)
35–49	31.3	(28.7–34.0)	45.2	(43.8–46.6)	44.4	(44.1–44.7)
50–64	50.9	(48.1–53.8)	33.3	(32.0–34.7)	30.7	(30.4–31.0)
Economic difficulties						
Some or many	10.5	(8.5–13.0)	42.2	(40.4–44.1)	49.8	(49.3–50.1)
None	89.5	(87.0–91.6)	57.8	(55.9–59.6)	50.3	(49.9–50.7)
Educational level						
Low	–	(–)	10.1	(9.0–11.2)	28.5	(28.2–28.9)
High	100.0	(–)	89.9	(88.8–91.0)	71.5	(71.1–71.8)
Geographical area						
North	39.3	(35.8–42.9)	43.4	(41.7–45.1)	41.6	(41.5–41.8)
Center	22.4	(19.9–25.1)	21.7	(20.4–23.0)	23.9	(23.7–24.0)
South and Islands	38.3	(34.5–42.3)	34.9	(33.1–36.8)	34.5	(34.4–34.6)
Urbanization Degree (population density)						
High	48.9	(45.2–52.7)	35.1	(33.3–37.0)	33.7	(33.4–34.0)
Intermediate	41.8	(38.1–45.6)	47.6	(45.8–49.3)	47.6	(47.2–47.9)
Low	9.3	(7.7–11.0)	17.3	(16.2–18.5)	18.7	(18.5–19.0)
Marital status						
Married	62.3	(58.6–65.8)	57.7	(55.9–59.5)	58.2	(57.8–58.6)
Not married	37.7	(34.2–41.4)	42.3	(40.5–44.1)	41.8	(41.4–42.2)
Nationality						
Italian	99.2	(98.3–99.6)	97.3	(96.8–97.8)	94.8	(94.6–95.0)
Others	0.8	(0.4–1.7)	2.7	(2.2–3.2)	5.2	(5.0–5.4)
Chronic diseases *						
None	86.9	(84.4–89.0)	84.7	(83.3–85.8)	86.1	(85.8–86.4)
At least one	13.1	(11.0–15.6)	15.4	(14.2–16.7)	13.9	(13.6–14.2)
Obesity						
Yes	6.4	(4.8–8.6)	7.9	(7.0–8.9)	9.4	(9.1–9.6)
No	93.6	(91.4–95.2)	92.1	(91.1–93.0)	90.6	(90.4–90.9)
Tobacco smoking						
Yes	16.4	(13.5–19.8)	25.4	(23.8–27.0)	28.7	(28.3–29.1)
No	83.6	(80.2–86.5)	74.6	(73.0–76.2)	71.3	(70.9–71.7)
Self-perceived health status						
Good or very good	79.3	(75.8–82.4)	74.4	(72.8–75.9)	75.7	(75.3–76.0)
Fair, bad, and very bad	20.7	(17.6–24.2)	25.6	(24.1–27.2)	24.3	(24.0–24.7)

* One or more of the following: kidney failure, chronic bronchitis, emphysema, respiratory failure, bronchial asthma, stroke or cerebral ischemia, diabetes, myocardial infarction, ischemic heart or coronary artery disease, other heart diseases, cancers (including leukemia and lymphoma), chronic liver disease or cirrhosis.

**Table 2 vaccines-11-01223-t002:** Adherence to seasonal influenza vaccination campaigns and Crude Odds Ratio (Crude OR) by socio-demographic characteristics among Medical Doctors (MDs), Non-Medical Health Personnel (NMHP), and General population excluding HP (NHP). PASSI 2014–2018 (*n* = 103,901).

	Healthcare Personnel (HP)					
	Medical Doctors (MDs) (*n* = 1178)		Non-Medical Health Personnel (NMHP) (*n* = 4801)		General Population (Excluding HP) (NHP) (*n* = 97,922)	
	% (IC 95%)	Crude OR (IC 95%)	% (IC 95%)	Crude OR (IC 95%)	% (IC 95%)	Crude OR (IC 95%)
Total	22.8 (19.8–26.1)	–	8.5 (7.5–9.5)	–	6.3 (6.1–6.5)	–
Gender						
Men	24.1 (20.0–28.8)	–	10.2 (8.3–12.4)	–	6.8 (6.5–7.0)	–
Women	21.3 (17.1–26.1)	0.852 (0.581–1.249)	7.8 (6.7–9.0)	0.742 * (0.566–0.974)	5.6 (5.3–5.9)	0.818 * (0.764–0.876)
Age group						
18–34	9.9 (6.2–15.3)		4.4 (3.2–5.9)		3.4 (3.1–3.8)	–
35–49	20.3 (15.6–26.1)	2.317 * (1.262–4.254)	6.4 (5.3–7.8)	1.507 * (1.09–2.209)	4.9 (4.6–5.2)	1.451 * (1.300–1.626)
50–64	28.4 (23.8–33.4)	3.596 * (2.019–6.404)	13.9 (11.8–16.3)	3.534 * (2.437–5.125)	10.6 (10.1–11.0)	3.336 * (2.991–3.720)
Economic difficulties						
Some or many	20.5 (13.2–30.6)	–	8.5 (6.9–10.4)	–	6.4 (6.1–6.7)	–
None	23.0 (19.8–26.6)	1.157 (0.664–2.015)	8.4 (7.3–9.6)	0.989 (0.755–1.295)	6.2 (5.9–6.5)	0.969 (0.905–1.038)
Educational level **						
Low	–	–	7.7 (5.4–10.9)		7.0 (6.6–7.4)	–
High	22.8 (19.8–26.1)	–	8.5 (7.5–9.7)	1.122 (0.750–1.679)	6.0 (5.7–6.2)	0.840 * (0.780–0.904)
Geographic area						
North	25.7 (21.2–30.8)		7.9 (6.7–9.2)		5.7 (5.5–6.0)	–
Center	17.9 (13.6–23.1)	0.629 * (0.417–0.948)	7.8 (6.1–9.8)	0.987 (0.731–1.333)	6.3 (6.0–6.7)	1.108 * (1.028–1.194)
South and Islands	22.7 (17.3–29.2)	0.848 (0.538–1.335)	9.6 (7.7–12.0)	1.248 (0.929–1.676)	6.9 (6.5–7.4)	1.227 * (1.131–1.331)
Urbanization Degree (population density)						
High	19.2 (15.3–23.9)	–	8.8 (7.1–10.7)	–	6.7 (6.4–7.1)	–
Intermediate	26.2 (21.2–31.9)	1.493 * (1.006–2.216)	7.8 (6.6–9.2)	0.883 (0.669–1.165)	6.0 (5.8–6.3)	0.891 * (0.824–0.964)
Low	26.2 (19.3–34.5)	1.494 (0.934–2.388)	8.7 (6.9–11.0)	0.994 (0.711–1.388)	6.1 (5.7–6.5)	0.901 * (0.821–0.988)
Marital status						
Married	25.0 (21.1–29.3)	–	9.1 (7.8–10.6)		6.9 (6.7–7.2)	–
Not married	19.1 (14.7–24.5)	0.709 (0.493–1.020)	7.6 (6.3–9.2)	0.826 (0.635–1.073)	5.4 (5.1–5.6)	0.759 * (0.707–0.815)
Nationality						
Italian	22.8 (20.0–26.4)	–	8.6 (7.6–9.7)	–	6.3 (6.1–6.5)	–
Others	–	–	4.9 (1.9–11.6)	0.543 (0.209–1.411)	5.5 (4.8–6.2)	0.862 * (0.754–0.986)

* Significant (*p* < 0.05); ** The logistic model for the medical doctors’ group does not include the educational level in the covariates.

**Table 3 vaccines-11-01223-t003:** Adherence to seasonal influenza vaccination campaigns and Crude Odds Ratio (Crude OR) by health conditions among Medical Doctors (MDs), Non-Medical Health Personnel (NMHP), and General population excluding HP (NHP). PASSI 2014–2018 (*n* = 103,901).

	Healthcare Personnel (HP)					
	Medical Doctors (MDs) (*n* = 1178)		Non-Medical Health Personnel (NMHP) (*n* = 4801)		General Population (Excluding HP) (NHP) (*n* = 97,922)	
	% (IC 95%)	Crude OR (IC 95%)	% (IC 95%)	Crude OR (IC 95%)	% (IC 95%)	Crude OR (IC 95%)
Total	22.8 (19.8–26.1)	–	8.5 (7.5–9.5)	–	6.3 (6.1–6.5)	–
Chronic disease **						
None	20.8 (17.6–24.3)	–	7.5 (6.5–8.6)		4.4 (4.3–4.6)	–
At least one	36.3 (27.9–45.7)	2.180 * (1.407–3.377)	13.9 (11.0–17.3)	1.996 * (1.481–2.691)	17.6 (16.8–18.5)	4.601 * (4.275–4.953)
Obesity						
Yes	30.0 (15.9–44.1)	1.493 (0.741–3.009)	10.6 (7.0–14.2)	1.312 (0.878–1.962)	9.2 (8.4–10.0)	1.586 * (1.428–1.762)
No	22.3 (19.1–25.5)	–	8.3 (7.2–9.3)	–	6.0 (5.8–6.2)	–
Tobacco smoking						
Yes	20.0 (13.9–27.9)	0.814 (0.505–1.311)	8.0 (6.2–10.3)	0.935 (0.682–1.281)	5.2 (4.9–5.6)	0.764 * (0.705–0.828)
No	23.5 (20.2–27.2)	–	8.6 (7.5–9.8)	–	6.7 (6.5–7.0)	–
Self–perceived health status						
Good or very good	21.3 (18.2–24.7)	0.672 (0.423–1.66)	7.6 (6.6–8.9)	0.691 * (0.524–0.910)	4.8 (4.6–5.0)	0.418 * (0.389–0.449)
Fair, bad, and very bad	28.7 (20.9–37.9)	–	10.7 (8.8–13.0)	–	10.8 (10.3–11.4)	

* Significant (*p* < 0.05); ** One or more of the following: kidney failure, chronic bronchitis, emphysema, respiratory failure, bronchial asthma, stroke or cerebral ischemia, diabetes, myocardial infarction, ischemic heart or coronary artery disease, other heart diseases, cancers (including leukemia and lymphoma), chronic liver disease or cirrhosis.

**Table 4 vaccines-11-01223-t004:** Adjusted Odds Ratio (Adj OR) of adherence to the seasonal influenza vaccination campaigns by socio-demographic characteristics and health conditions among Medical Doctors (MDs), Non-Medical Health Personnel (NMHP), and General population excluding HP (NHP). PASSI 2014–2018 (*n* = 103,901).

	Healthcare Personnel (HP)					
	Medical Doctors (MDs) (*n* = 1178)		Non-Medical Health Personnel (NMHP) (*n* = 4801)		General Population (excluding HP) (NHP) (*n* = 97,922)	
	AdjOR	(IC 95%)	AdjOR	(IC 95%)	AdjOR	(IC 95%)
Gender						
Men	–	–	–	–	–	–
Women	1.014	(0.662–1.554)	0.741 *	(0.557–0.986)	0.808 *	(0.752–0.867)
Age group						
18–34	–	–	–	–	–	–
35–49	2.223 *	(1.195–4.138)	1.518 *	(1.020–2.261)	1.358 *	(1.210–1.524)
50–64	3.135 *	(1.681–5.846)	3.299 *	(2.216–4.912)	2.473 *	(2.196–2.785)
Educational level **						
Low					–	–
High					1.069	(0.988–1.158)
Geographic area						
North	–	–	–	–	–	–
Center	0.644 *	(0.422–0.983)	0.967	(0.707–1.321)	1.097 *	(1.014–1.187)
South and Islands	0.792	(0.492–1.276)	1.197	(0.885–1.620)	1.229 *	(1.129–1.338)
Urbanization Degree (population density)						
High	–	–			–	–
Intermediate	1.415	(0.954–2.097)			0.907 *	(0.835–0.984)
Low	1.428	(0.882–2.313)			0.899 *	(0.816–0.991)
Marital status						
Married	–	–	–	–	–	–
Not married	0.951	(0.636–1.422)	1.039	(0.794–1.360)	1.108 *	(1.026–1.196)
Nationality						
Italian					–	–
Others					1.092	(0.949–1.256)
Chronic disease **						
None	–	–	–	–	–	–
At least one	1.852 *	(1.169–2.934)	1.729 *	(1.268–2.357)	3.490 *	(3.225–3.775)
Obesity						
Yes			1.115	(0.737–1.688)	1.096	(0.980–1.227)
No			–	–	–	–
Tobacco smoking						
Yes					0.768 *	(0.706–0.834)
No					–	–
Self–perceived health status						
Good or very good	0.859	(0.529–1.394)	0.982	(0.726–1.330)	0.685 *	(0.633–0.742)
Fair, bad, and very bad	–	–	–	–	–	–

* Significant (*p* < 0.05); ** One or more of the following: kidney failure, chronic bronchitis, emphysema, respiratory failure, bronchial asthma, stroke or cerebral ischemia, diabetes, myocardial infarction, ischemic heart or coronary artery disease, other heart diseases, cancers (including leukaemia and lymphoma), chronic liver disease or cirrhosis.

## Data Availability

The data presented in this study are available on request from the corresponding author. The data are not publicly available due to restrictions, e.g., privacy or ethical.

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
