# Peer review of "Influenza Vaccination Uptake and Prognostic Factors among Health Professionals in Italy: Results from the Nationwide Surveillance PASSI 2015–2018"

_vaccines, 2023, doi:10.3390/vaccines11071223_

Round 1

Reviewer 1 Report

This paper presents the results of large scale high quality survey on influenza vaccine adoption in Italy.   The results mainly focus on vaccination rates in the medical profession, and compares these rates to those in the general population. 

The presented results are interesting, and maybe counter intuitive:

1. Given the health impact on patients, I expected doctors to have mandatory flu vaccination.  If this is not the case, please say so in the paper.  Even so, a ~20% vaccination rate seems low.  It may be that the recommendation is to get vaccinated ever other or third year. Again, if this is the case, please indicate it in the text.   If the recommendation is to get vaccinated every other year, I would suggest to analyze two years of data to investigate adherence to health guidelines.

2. I would have expected women to be more likely to be vaccinated, as they tend to have more contacts with the health care system. 

3.  I would have expected individuals with higher education level to have a higher vaccination rate.  This was not the case, I am hard pressed to explain this. 

From a methodology point of view, it may be interesting (but not required for this manuscript) to correct for possible survey response biases by using statistical analysis methods that use propensity scores. 

Author Response

This paper presents the results of large scale high quality survey on influenza vaccine adoption in Italy.   The results mainly focus on vaccination rates in the medical profession, and compares these rates to those in the general population. 

The presented results are interesting, and maybe counter intuitive:

  1. Given the health impact on patients, I expected doctors to have mandatory flu vaccination.  If this is not the case, please say so in the paper.  Even so, a ~20% vaccination rate seems low.  It may be that the recommendation is to get vaccinated ever other or third year. Again, if this is the case, please indicate it in the text.   If the recommendation is to get vaccinated every other year, I would suggest to analyze two years of data to investigate adherence to health guidelines.

As indicated at lines 66-67, in Italy flu vaccination is recommended to health professionals. Within the discussion paragraph “4.1.2. Interventions and application of proven effective policies”, the article proposes a specific part (lines 318-329) on the unsolved debate around mandating vaccination among healthcare providers.

  1. I would have expected women to be more likely to be vaccinated, as they tend to have more contacts with the health care system. 

Thanks for sharing an expectation of yours about the vaccination attitude according to gender. We have reported what resulted from the multivariate analysis, and likewise also other Authors in their studies found out this discrepancy, such as Amodio et al., 2010 or Amodio et al., 2011b; Barbadoro et al., 2020; Bianchi et al., 2021; Bonaccorsi et al., 2013 or Bonaccorsi et al., 2015; Maffeo et al., 2020; Rabensteiner et al., 2018 (reference to Table 1 enclosed as supplementary file).

  1. I would have expected individuals with higher education level to have a higher vaccination rate.  This was not the case, I am hard pressed to explain this. 

Concerning the role of sociodemographic variables such as education level in predicting influenza vaccination uptake, scientific literature reported contrasted results, mostly depending on the country in which the studies have been conducted and on the target groups. In particular, some Authors have described a positive association between social level and vaccination coverage (the lower the social level, the lower the influenza vaccination uptake) while others have found no association. As suggested by Lucyk et al., in countries with universal, publicly-funded health care systems, as is the case of Italy, the burden exerted by social level including education on influenza vaccination uptake is little or absent, since it is reduced by an equitable access, free of charge, for all the categories provided by law.

[Vukovic, V.; Lillini, R.; Lupi, S.; Fortunato, F.; Cicconi, M.; Matteo, G.; Arata, L.; Amicizia, D.; Boccalini, S.; Bechini, A.; et al. Identifying people at risk for influenza with low vaccine uptake based on deprivation status: A systematic review. Eur. J. Public Health. 2020, 30, 132–141.

Lucyk, K.; Simmonds, K.A.; Lorenzetti, D.L.; Drews, S.J.; Svenson, L.W.; Russell, M.L. The Association between Influenza Vaccination and Socioeconomic Status in High Income Countries Varies by the Measure Used: A Systematic Review. BMC Med. Res. Methodol. 2019, 19, 153.

Jain, A.; van Hoek, A.J.; Boccia, D.; Thomas, S.L. Lower Vaccine Uptake amongst Older Individuals Living Alone: A Systematic Review and Meta-Analysis of Social Determinants of Vaccine Uptake. Vaccine 2017, 35, 2315–2328.

Nagata, J.M.; Hernández-Ramos, I.; Kurup, A.S.; Albrecht, D.; Vivas-Torrealba, C.; Franco-Paredes, C. Social Determinants of Health and Seasonal Influenza Vaccination in Adults ≥65 Years: A Systematic Review of Qualitative and Quantitative Data. BMC Public Health 2013, 13, 388.].

From a methodology point of view, it may be interesting (but not required for this manuscript) to correct for possible survey response biases by using statistical analysis methods that use propensity scores.          

The PASSI surveillance system refers to a cross-sectional study design which, basing on the US CDC model of Behavioral Risk Factor Surveillance Systems, has been ensuring very high response rates over the years. The scientific article “Baldissera, S.; Ferrante, G.; Quarchioni, E.; Minardi, V.; Possenti, V.; Carrozzi, G.; et al., Field substitution of nonresponders can maintain sample size and structure without altering survey estimates-the experience of the Italian behavioral risk factors surveillance system (PASSI), Ann. Epidemiol. 2014, 24(4), 241–245.” clarifies the mechanism in place for field-substitution technique, as well as other standard procedures both in the data collection and monitoring process which are in place to check, control and minimise the possible response biases.

Reviewer 2 Report

This important manuscript by Minardi and colleagues aims to provide Italian estimates on influenza vaccination coverage rates among health professionals by using data from an Italian Surveillance System. It investigates the association between demographic, medical, and socio-economic factors and adherence to immunization programs for influenza among healthcare professionals.

I want to congratulate the authors on this scientifically sound work, which addresses an important issue that has not yet found a solution.

Please find below some comments, suggestions, and doubts:

- I am struggling to understand why the authors have analyzed and reported (but not discussed, actually) data from a sample of non-healthcare professionals (general population excluding HP). In my opinion, the three groups are not comparable at all (the authors specify that influenza vaccination is not recommended for this third group), and the juxtaposition of this group with the others may be misleading for readers. Moreover, the comparison of this group with the others (or the reason why the results are shown) is not reported or mentioned in the aim of the study or discussed in the discussion section. I feel that this group should not be included in the study. If the authors disagree, I suggest revising and integrating the manuscript (e.g., in the aim and discussion) to provide a clear picture to the reader of the added value of this comparison.

- In the discussion, I suggest emphasizing the need for a national registry of vaccination, which would change the landscape of vaccination coverage monitoring in Italy.

- Does PASSI collect data on the setting in which each HCP works? It would be interesting to have a description (and comparison) of different groups (e.g., HCP working in hospitals, nursing homes, primary care services, etc.) to understand whether there are significant differences in influenza vaccination uptake. This would be a novel addition and provide important information for this manuscript.

- Are there data on the level of health/vaccine literacy of the sample? It would be interesting to know whether there are differences between MDs and NMHP and whether higher levels of HL/VL are associated with higher influenza vaccination uptake, as this is still quite unclear (1-2)

- Minor: Line 274 - Please provide a better explanation of what is meant by the "vaccine confidence gap.

1. https://pubmed.ncbi.nlm.nih.gov/35682272/

2. https://pubmed.ncbi.nlm.nih.gov/35098752/

Author Response

This important manuscript by Minardi and colleagues aims to provide Italian estimates on influenza vaccination coverage rates among health professionals by using data from an Italian Surveillance System. It investigates the association between demographic, medical, and socio-economic factors and adherence to immunization programs for influenza among healthcare professionals.

I want to congratulate the authors on this scientifically sound work, which addresses an important issue that has not yet found a solution.

Please find below some comments, suggestions, and doubts:

- I am struggling to understand why the authors have analyzed and reported (but not discussed, actually) data from a sample of non-healthcare professionals (general population excluding HP). In my opinion, the three groups are not comparable at all (the authors specify that influenza vaccination is not recommended for this third group), and the juxtaposition of this group with the others may be misleading for readers.

Moreover, the comparison of this group with the others (or the reason why the results are shown) is not reported or mentioned in the aim of the study or discussed in the discussion section. I feel that this group should not be included in the study. If the authors disagree, I suggest revising and integrating the manuscript (e.g., in the aim and discussion) to provide a clear picture to the reader of the added value of this comparison.

Thanks for commenting on this inconsistency that we hope now is better explained. As suggested, in fact, in different parts (Introduction: lines 71-76, Methods: lines 79-82: Discussion: lines 245-247) we have entered the clarification that, starting from population-based data, it has been possible to isolate the information concerning the flu vaccination uptake in health professionals (doctors vs others) vs. the individuals who are not engaging in the healthcare sector. Given very low rates in the general population, indeed the article focuses mainly on health personnel because they are targeted by annual recommendation in the light of their increased risk exposure.

- In the discussion, I suggest emphasizing the need for a national registry of vaccination, which would change the landscape of vaccination coverage monitoring in Italy.

Thanks for making us reason on this issue that maybe was not very clearly assessed in the article. We have improved in defining better the state of the art in matter of regional/national vaccine registries. This clarification is made in the Introduction (lines: 69-70) and recalled in the Discussion (lines: 312-317).

- Does PASSI collect data on the setting in which each HCP works? It would be interesting to have a description (and comparison) of different groups (e.g., HCP working in hospitals, nursing homes, primary care services, etc.) to understand whether there are significant differences in influenza vaccination uptake. This would be a novel addition and provide important information for this manuscript.

Unfortunately, PASSI does not collect data on the specific healthcare setting. We have clearly indicated in the Limitations’ section. Thanks for this comment.

- Are there data on the level of health/vaccine literacy of the sample? It would be interesting to know whether there are differences between MDs and NMHP and whether higher levels of HL/VL are associated with higher influenza vaccination uptake, as this is still quite unclear (1-2)

No, this kind of information is not available. Just as example, basing on PASSI data, in a specific Region, it was possible to investigate Health Literacy, Socioeconomic Status and Influenza Vaccination Uptake (https://doi.org/10.3390/ijerph19116925).

- Minor: Line 274 - Please provide a better explanation of what is meant by the "vaccine confidence gap.

  1. https://pubmed.ncbi.nlm.nih.gov/35682272/

Lorini C, Lastrucci V, Zanella B, Gori E, Chiesi F, Bechini A, Boccalini S, Del Riccio M, Moscadelli A, Puggelli F, Berti R, Bonanni P, Bonaccorsi G. Predictors of Influenza Vaccination Uptake and the Role of Health Literacy among Health and Social Care Volunteers in the Province of Prato (Italy). Int J Environ Res Public Health. 2022, 19(11), 6688. doi: 10.3390/ijerph19116688.

  1. https://pubmed.ncbi.nlm.nih.gov/35098752/

Ramachandran S, Shuvo SA, Behal M, Hagemann T, Hohmeier KC, Chiu C-Y. Social determinants of health and adult influenza vaccination: a nationwide claims analysis. 2022, Journal of Managed Care & Specialty Pharmacy, 28 (2), 196-205. doi: 10.18553/jmcp.2022.28.2.196.

Thanks for having indicated these two further references, we have entered a further explanation of the vaccine confidence gap as well described in the Vaccine Confidence Report that we considered among the relevant information sources on vaccination-related issues.

Reviewer 3 Report

This is an overall interesting ms by V. Mirandi et al on influenza vaccination uptake among health professionals in Italy. 

There were some difficulties in the comprehension of the text mainly due to syntactical and grammatical errors. 

In general the use of glossological manners of speech should be avoided.

Therefore I strongly suggest the proofreading by a native English speaker. 

Introduction

The introductory part was repetitive in many parts while the main points could get across in a less lengthy manner.

(42-43): underlying chronic health conditions, even comorbidities (they are the same)

(45) vaccine preventable disease (the syntax in this phrase fails to convey what the author has in mind

(46-50) In fact,… (needs to be rephrased to be more clear)

(63-66) please rephrase to be more clear

(71) no need to repeat “In Italy”

Materials and Methods

2.1 can be included in 2.1.1

(93) LHU should be better explained

(112) why did the authors chose to include only employed participants? The reasoning behind this decision should be clear

(119) consider: do you mean include?

(132) in making ends meet:such figures of speech are usually avoided

(135) marital status just defined as married and unmarried excludes other categories (divorced, in a relationship etc)

In the health related conditions the authors do not include immunosuppression of any kind. Vaccination of mmunosuppressed patients is usually a priority in most national guidelines 

Results

There are grammatical errors which substract from the value of the research

(189) an obesity condition : could be replaced by “obesity”

(209) >35: probably the authors mean <35

(211) cultural level: do you mean socioeconomic status?

(216) Additionally,… this phrase is not clear

Table 2a no results in non Italians MDs (although a small number)

(231) A quite strong finding…. Please rephrase to make it clear

Table 2b crude OR 2,180 for chronic disease in MDs is not in concordance with difference in %.

(248) AdjOR 1,85 almost one: do you mean at least one?

Discussion 

The authors compare people who practise health profession to other professionals. They do not discuss the fact that vaccination of the general population is not widely advices.

(274) what is “vaccine confidence gap”?

(278) where is table S1

(282) anything seems to be changed: you mean nothing?

(318) PNPV should be defined

(324) In other countries: generalisation 

(337) Many HP…. Please rephrase, not clear

(This is not a sentence)

(351) occurance?

(369-371) please rephrase. Hard to comprehend 

The word basing is been used numerous times throughout the text which is grammatically incorrect and repetitive. 

(380)survey not surveys

(393) the word unusual should be deleted

(402) reconstructed retrospectively: it is mentioned for the first time in the text. Should this be included in the methods

The authors do not discuss the differences of vaccination coverage among different areas and the fact that MDs were older than the rest of the participants

You should consider making meticulous corrections and changes in the discussion part.

The references include mainly Italian studies

The comments are included in my review

Author Response

This is an overall interesting ms by V. Mirandi et al on influenza vaccination uptake among health professionals in Italy. There were some difficulties in the comprehension of the text mainly due to syntactical and grammatical errors. In general the use of glossological manners of speech should be avoided.

Therefore I strongly suggest the proofreading by a native English speaker. 

Introduction

The introductory part was repetitive in many parts while the main points could get across in a less lengthy manner.

Thanks for commenting on this, we have simplified and eliminated all the redundancies.

(42-43): underlying chronic health conditions, even comorbidities (they are the same)

Thanks for checking, we have remarked the presence of a chronic condition or the comorbidity condition.

(45) vaccine preventable disease (the syntax in this phrase fails to convey what the author has in mind

Thanks, we have moved the VPD reference because it failed to convey the intended meaning.

(46-50) In fact, (needs to be rephrased to be more clear)

Thanks, we have modified.

(63-66) please rephrase to be more clear

Thanks, we have rephrased.

(71) no need to repeat “In Italy”

Thanks, we have eliminated.

Materials and Methods

2.1 can be included in 2.1.1

Thanks, we have incorporated.

(93) LHU should be better explained

Thanks, we have strengthened the definition.

(112) why did the authors chose to include only employed participants? The reasoning behind this decision should be clear.

Thanks, we have better explained, by introducing the sample paragraph with the explanation of this target group selection.

(119) consider: do you mean include?

Thanks, we have modified.

(132) in making ends meet:such figures of speech are usually avoided.

Please, check the Eurostat indicator table that indicates this exact technical definition as enclosed in the Word file.

PASSI uses it since its establishment as surveillance system in 2007.

(135) marital status just defined as married and unmarried excludes other categories (divorced, in a relationship etc)

Thanks for you note, yes we confirm that the dichotomous variable is about being married or not.

In the health related conditions the authors do not include immunosuppression of any kind. Vaccination of immunosuppressed patients is usually a priority in most national guidelines.

Thanks for this relevant comment, anyway PASSI does not include immunosuppression conditions.

Results

There are grammatical errors which substract from the value of the research

We carefully check also with the contribution of a native English speaker.

(189) an obesity condition : could be replaced by “obesity”

Thanks, we have modified.

(209) >35: probably the authors mean <35

Thanks, we have modified.

(211) cultural level: do you mean socioeconomic status?

Thanks, we have modified with the correct variable that is education level.

(216) Additionally,… this phrase is not clear

Thanks, we have modified.

Table 2a no results in non Italians MDs (although a small number)

Because of the very small number of non-Italian MDs, the analysis of this figure was not included. 

(231) A quite strong finding…. Please rephrase to make it clear

Thanks, we have modified.

Table 2b crude OR 2,180 for chronic disease in MDs is not in concordance with difference in %.

Thanks you for your point, we checked results and can confirm they are correct.

In case of persisting doubts about this figure, may we ask some clarifications about this issue.

(248) AdjOR 1,85 almost one: do you mean at least one?

Thanks, we have modified.

Discussion 

The authors compare people who practise health profession to other professionals. They do not discuss the fact that vaccination of the general population is not widely advices.

Thanks, we have better explained the overall rationale of the study and its focus on HP, justifying the selection of this crucial target group.

(274) what is “vaccine confidence gap”?

Thanks, we have better explained the concept that we referred to mostly in the Discussion.

(278) where is table S1

Table S1 is a ponderous attachment that we have annexed in the Supplementary file (this is the reason for the labelling “S1” as required by the journal rules). This table includes all the relevant studies on flu vaccination among health professionals conducted in Italy in the last three decades. This explanation addresses also the answer to the last comment.

(282) anything seems to be changed: you mean nothing?

Thanks, we have changed accordingly.

(318) PNPV should be defined

Thanks, we have strengthened its definition, role and principles.

(324) In other countries: generalisation 

Thanks, we have modified accordingly.

(337) Many HP…. Please rephrase, not clear (This is not a sentence)

Thanks, we have rephrased.

(351) occurance?

Thanks, we have changed.

(369-371) please rephrase. Hard to comprehend 

Thanks, we have rephrased.

The word basing is been used numerous times throughout the text which is grammatically incorrect and repetitive. 

Thanks, we have checked and changed the use of “basing” everywhere.

(380) survey not surveys

Thanks, we have corrected.

(393) the word unusual should be deleted

Thanks, we have deleted.

(402) reconstructed retrospectively: it is mentioned for the first time in the text. Should this be included in the methods

Thanks, we have better explained the methodological process that is starting from PASSI data we operated the selection of working-age individuals and then health professionals.

The authors do not discuss the differences of vaccination coverage among different areas and the fact that MDs were older than the rest of the participants

Thank you for your suggestion, we added some points in the discussion as follow.

The differences of vaccination coverage observed among different areas (better rates in the North and in the South than in the Center) may reflect indirectly the regional policies.

When considering prevalences by age in the different HP categories, MDs show higher rates than in NMHP in all age groups. (Please, consider this further table that supports the statement about how MDs behave in the different age groups).

gruppi

età

%

IC95%

MD

18-34

9.9%

6.2%

15.4%

35-69

25.5%

22.0%

29.3%

NMHP

18-34

4.4%

3.2%

5.9%

35-69

9.5%

8.4%

10.8%

You should consider making meticulous corrections and changes in the discussion part.

We do hope that the reviewer will appreciate and acknowledge the intensive modification process that the article has undergone overall.

The references include mainly Italian studies

Please, consider the explanation given with reference to Table S1.

Round 2

Reviewer 2 Report

Many thanks to the authors for addressing all the comments. I have no further suggestions.